# Bullous Pemphygoid and Novel Therapeutic Approaches

**DOI:** 10.3390/biomedicines10112844

**Published:** 2022-11-08

**Authors:** Giovanni Marco D’Agostino, Giulio Rizzetto, Andrea Marani, Samuele Marasca, Matteo Candelora, Daisy Gambini, Helena Gioacchini, Edoardo De Simoni, Andrea Maurizi, Anna Campanati, Annamaria Offidani

**Affiliations:** Dermatological Clinic, Department of Clinical and Molecular Sciences, Polytechnic Marche University, 60126 Ancona, Italy

**Keywords:** bullous pemphigoid, target therapies, biologics, small molecules, novel treatments

## Abstract

Bullous pemphigoid is a subepidermal blistering disease associated with autoantibodies (auto-ab) to BP180 and BP230 which affects elderly patients, predominately. Although it is a rare disease, bullous pemphigoid is the most common among the autoimmune bullous skin diseases. Systemic corticosteroids and immunosuppressants represent milestones in the treatment of patients suffering from bullous pemphigoid; however, therapeutic management of patients still represents a clinical challenge, owing to the chronic nature of the disease and to potential adverse effects related to the long-term use of systemic treatments. Recent discoveries on the pathogenesis of bullous pemphigoid have allowed investigation of new target therapies against selective pro-inflammatory mediators. These therapies appear to yield satisfactory results with fewer side effects in cases of refractory disease. The review discusses current evidence on these new therapeutic targets and specific drugs under investigation.

## 1. Introduction

Bullous pemphigoid (BP) is a rare autoimmune skin disease which affects the elderly in the eighth decade of life predominantly [1].

In older populations, multiple coexisting comorbidities and exposure to drugs able to potentially trigger the disease give reasons for the increase of BP incidence in recent years, ranging in Europe from 2.5 to 42.8 cases/million/year [2].

Classic BP is characterized by tense bullous lesions on normal or erythematous/edematous skin and intense itching, mainly located on the groin and axillary folds, the thigh, and the lower abdomen. Furthermore, oral, genital, or esophageal mucosal lesions are involved in 10–20% of cases [3].

The pathogenesis of BP has been identified as the production of autoantibodies against the hemidesmosome antigens BP180 and BP230, leading to a detachment at the dermo-epidermal junction. It is reported in the literature that levels of disease activity correlate with the circulating titers of anti-BP180 IgG and IgE antibodies [4]. IgE promotes the local infiltration of eosinophils, leading to the formation of bullae by two mechanisms. First, anti-BP180 IgE may bind to the FcεRI receptors on mast cells, leading to a cross-link with the hemidesmosome, degranulation, and histamine release, amplifying the chemotaxis of eosinophils and neutrophils. Secondly, IgE may directly bind to BP180 on keratinocytes, be internalized, and stimulate the release of interleukin (IL) 6 and IL-8, with a chemotactic effect [5].

BP may be associated with various disorders. A systematic review associates BP with a possible increase in hematological malignancies, although no statistically increased overall risk of developing a malignancy has been identified in BP patients [6]. It has been shown that BP may increase thrombotic risk, being a disease mediated by Th1 and Th2 cells, producing inflammatory cytokine cascades and inducing an upregulation of vascular endothelial growth factor and E-selectin, which promotes endothelial cell activation [7].

The prognosis of BP has been evaluated in a meta-analysis showing a 1-year combined mortality rate of 23.5%, and superinfection of skin ulcers is a leading cause of death [8].

Therapy is challenging, as it is based on the use of systemic steroids to induce remission, followed by tapering the dose slowly while trying to prevent new bullae from forming. Because BP is a chronic disease, therapy will have a long duration, and the side effects of chronic steroid intake may occur. Other canonical therapies include drugs defined as steroid-sparing, such as azathioprine, methotrexate, mycophenolate mofetil, dapsone, tetracyclines, and intravenous immunoglobulins [9].

However, the problem of BP being refractory to conventional therapies is the reason that prompted us to carry out a literature review with the purpose of analyzing the different treatment options available and considering some new therapies, in particular biologics.

## 2. Materials and Methods

This scoping review was based on the approach developed by Arksey and O′Malley that includes five essential steps: identification of the research question; identification of appropriate studies; selection of studies; tracking of data; and collection, summarization, and reporting of results. The Preferred Reporting Items for Systematic Reviews and Meta-Analysis (PRISMA) extension for scoping review criteria was used to guide the conduction and reporting of the review [10].

### 2.1. Identification of the Research Question

A brainstorming approach involving the entire research team was used to identify the research questions. The research group included six dermatologists with expertise in the research field of bullous diseases and clinical management of patients.

At the initial meeting, the group identified the research question and determined the research strategy. The research question was: “which novel therapeutic approaches have been/are emerging in the last 10 years for management of patients with bullous pemphigoid?”

### 2.2. Study Selection Process

We performed a worldwide systematic review of studies reporting on bullous pemphigoid, using 3 electronic medical databases–PubMed, EMBASE, and Web of Science—and considering articles dated 1 January 2012 to 1 May 2022.

The search terms were selected to identify studies describing novel therapeutic approach to pemphigoids.

The keywords used were “bullous pemphigoids AND novel treatments”, “bullous pemphigoids AND biologics”, “bullous pemphigoids AND small molecules”, and “bullous pemphigoids AND target therapies”.

All selected databases were searched from their respective inceptions. In addition, we searched by hand the reference lists of other relevant articles on therapeutic approaches to bullous pemphigoid. 

In this first phase, 100 records were identified from the selected databases. The number of records after duplicates were removed was 82. Among the selected records, none was marked as ineligible by automation tools. Relevant studies were then chosen. This process occurred in three phases. In the first phase, three researchers (GM.D, G.R., and A.Mar) independently selected articles based on their titles. Any disagreements were resolved by consulting a senior researcher (A.O.). In the second phase, abstracts were evaluated. Three members of the research team (GM.D, G.R., and A.Mar), independently evaluated each abstract. The research group resolved all discrepancies through unanimous consent. Twelve articles were excluded as not related to humans, and 70 were evaluated for full-text analysis. Among them, 12 manuscripts were not retrieved, and thus the documents assessed for eligibility numbered 58. 

The third phase consisted of critical appraisal of the full text of the 58 selected papers. To be included into our mini-review, studies had to be focused on novel therapeutic treatments for bullous pemphigoid, clinical course, and response to systemic therapies. All included studies had to be published in English, with the abstract available. No restrictions on study design were considered, and in vitro and in vivo pre-clinical trials, controlled clinical trials, case-control studies, cross-sectional studies, and case series were included. Articles were excluded from our review for three reasons only: reason 1 was articles not including therapeutic intervention (28 reports were excluded for this reason), reason 2 was articles being case reports (6 reports were removed for this reason), and reason 3 was reports being written in languages other than English (4 reports were excluded for this reason). 

### 2.3. Data Extraction

A data extraction module was designed by A.C. before data extraction to accelerate the entire process. To answer the research question, the following information was extracted from the included articles: Author(s) name and publication date; study design; study population; sample size; measured outcomes; study results; and study recommendations.

## 3. Results

The flowchart of the PRISMA study is shown in Figure 1. Our search identified 82 records after removing duplicates. After review of the titles and abstracts, 24 citations were dropped (research not related to humans or reports not retrieved), and 58 were evaluated for full-text eligibility. After review of the full text, 20 pre-clinical trial, controlled clinical trial, case-control study, cross-sectional study, case series, or review articles were found to be eligible and included in this study. 

The data found show that novel therapeutic approaches to bullous pemphigoid are emerging. 

### 3.1. Rituximab

Rituximab is a chimeric IgG1 monoclonal antibody that targets the CD20 receptor located at the surface of B-lymphocytes [11]. The mechanisms by which rituximab can result in B-lymphocyte depletion number at least four: antibody-dependent cellular cytotoxicity, antibody-dependent cell phagocytosis, complement-dependent cytotoxicity, and direct stimulation of cell apoptosis or other cell death mechanisms. In this way, rituximab prevents the differentiation of lymphocytes into antibody-secreting plasma cells. In addition, rituximab also modulates T lymphocyte activity by inhibiting CD4+ T lymphocytes and increases the number and functioning of FOXP3+ regulatory cells. 

In BP, several case reports and retrospective studies have demonstrated a good response, supporting its efficacy and safety [12]. The role of B-lymphocytes in the pathogenesis of bullous diseases encompasses several cellular functions including secretion of autoantibodies, aiding in T-cell activation, and production of pro-inflammatory cytokines. Therefore, B-lymphocytes are an important therapeutic target in these diseases, and selective depletion of B-lymphocytes (such as through the use of monoclonal antibodies) is a well-established therapy in the treatment of autoimmune bullous diseases. Nevertheless, prolonged depletion of B-lymphocytes may result in an increased risk of infection, although some studies seem to indicate that rituximab acts primarily on plasma cells responsible for the production of pathogenic autoantibodies and not on CD 20 plasma cells, which produce antimicrobial antibodies with protective functions. Rituximab also has an impact on antigen-specific T lymphocytes, although it does not go on to influence their overall number and functioning [13,14].

Nowadays, rituximab is approved for treatment of pemphigus, but its administration for BP is still off-label. 

The first report on the use of rituximab in BP is dated to 2007 and based on the lymphoma protocol, while now the rheumatoid arthritis (RA) protocol is used.

Generally, the same dosing schedule is used for BP as in pemphigus vulgaris: 1000 mg W0, followed by 1000 mg W2; thereafter, at months 12 and 18 and every 6 months after clinical reevaluation, a dose of 500 mg can be given. Premedication with antipyretics and antihistamines is mandatory to minimize the risk of post-infusive reactions [15,16].

The efficacy profile of rituximab in BP is variable based on different studies reviewed; overall, a complete response was achieved in 60–70% of cases, with a relapse occurring in 20% of cases. However, controlled clinical studies are lacking [17,18].

The first study analyzed is a retrospective case-control study performed between 2010 and 2012 in Taiwan and included patients with generalized BP who required systemic therapies. Therapy in the first group of patients (group R) included weekly administration of rituximab at a dose of 500 mg for 4 weeks and corticosteroids with a starting dose of prednisolone of 0.5 mg/kg daily. The corticosteroid dose was scaled up rapidly after disease control was achieved. Each dose had a duration of 3–4 weeks, with a total treatment duration less than 6 weeks. A second group of patients (group C) with similar disease severity receiving a similar starting dose of prednisolone for at least 6 months was considered.

More than 90% of patients in the R group achieved complete remission, considered as no new active lesion onset for at least 2 months. This percentage was significantly higher than in the control C group (*p* = 0.02) [19]. 

A second retrospective case-control study involving 13 patients reported complete remission in 90% of BP patients who received a combination of rituximab and prednisone. When compared with a second group of patients who received conventional immunosuppressive therapy, the first group showed lower rates of infection and mortality due to the earlier corticosteroid dose-tapering allowed by rituximab administration [20].

Rituximab was then used in combination with several drugs, particularly mycophenolate mofetil, azathioprine, methotrexate, doxycycline, and dapsone. Data from a 20-patient study showed that 75% of those patients achieved a durable response with rituximab, with 5 patients requiring adjuvant therapy, 7 requiring minimal therapies, and 3 no longer taking any therapy. In addition, nine patients were no longer taking prednisone at their last visit, suggesting steroid-sparing activity enabled by rituximab [21,22,23].

Rituximab has also been used in combination with immunoglobulins. In a retrospective study, the efficacy and safety of a protocol combining immunoglobulins and rituximab (at 375 mg/m^2^ for 12 infusions) was evaluated. All patients remained in remission with the absence of adverse events for 6 years. In addition, the authors reported no serious infections. In another retrospective study, 12 patients (with a mean age of 68.25 years) unresponsive to immunosuppressive therapy were treated with rituximab and immunoglobulins. Complete clinical resolution was achieved in an average of 6.4 months, and previous systemic therapy was discontinued in 6.2 months. Two patients relapsed after therapy but responded to further infusions of rituximab. The other 10 patients did not relapse. All patients remained in remission without adverse events for 6 years [24,25,26].

Rituximab has also been successfully used in the treatment of nivolumab-induced BP in combination with plasmapheresis. We report a case report of a 67-year-old male with stage IV melanoma and metastases to the liver and lung who was treated with nivolumab at a dose of 3 mg/kg every 2 weeks. After 16 cycles in 32 weeks, the patient developed a severe form of BP with 90% involvement of the skin surface and altered consciousness. A laboratory confirmed the diagnosis of BP with BP180 and BP230 antibody positivity. After a failure of systemic corticosteroid therapy, rituximab was used at a dose of 1000 mg in two administrations 15 days apart. This resulted in complete remission of the disease. The patient then continued therapy with a topical corticosteroid (betamethasone dipropionate) without the need for systemic therapy [27].

Rituximab, which was otherwise comparable to omalizumab, proved to be superior to omalizumab for relapse prevention. Rituximab showed a lower relapse rate and a longer time before relapse than omalizumab. A complete clinical response was achieved in 85% of cases with rituximab and 84% of cases with omalizumab. The relapse rate was significantly lower with rituximab (29%) than with omalizumab (80%) [28].

Therapy with rituximab results in a number of biochemical changes. The most prominent of these is a major reduction in B-lymphocytes, as confirmed by phenotypic analysis. A dramatic decrease in anti-BP180 antibodies and in the frequency of circulating BP180-specific B-cells was also found following therapy. This was accompanied by an improvement in skin manifestations. 

A significant change was then observed in the frequency of cytokines expressed in the BP180-specific B-cell population. In contrast, this change was not found in the BP180 negative B-cell population. In particular, a marked decrease in the expression of the pro-inflammatory cytokines IL-15 and IL-6 was found after treatment with rituximab. This, together with the increased frequency of the anti-inflammatory cytokines IL10 and IL1RA, might be involved in the long-term remission of BP patients after rituximab therapy [29].

In conclusion, judging from the studies reviewed, although rituximab appears to have good efficacy and safety in the treatment of BP, the lack of controlled clinical trials in the current state of the art prevents its use for non-responsive forms of BP in routine clinical practice. All the results described above are summarized in Table 1.

### 3.2. Dupilumab

Dupilumab is a fully human monoclonal antibody that binds to the alpha subunit of the IL-4 receptor. The alpha subunit is shared by both IL-4 and IL-13, two cytokines involved in the type 2 inflammatory pathway [30].

Patients with BP show greater concentrations of IL-4- and IL-13-producing CD4+ and CD8+ T cells in their blister fluids and sera than healthy people [31]. BP patients also have elevated blood levels of immunoglobulin E and eosinophilia [32].

IgE-mediated IL-4 and IL-13 production seems to be involved in the upregulation of Th2, the predominant immunological response in BP patients, indicating Th2 lymphocytes′ role in the loss of tolerance of BP180 [33]. Autoreactive Th2 cells are believed to play a double role in BP, on one hand by stimulating B-cells′ autoantibody production and on the other hand by participating in the recruitment and activation of eosinophils, which contributes to maintaining a Th2-type inflammatory response by producing IL-4, IL-5, and IL-13. Therefore, it has been hypothesized that dupilumab indirectly exerts its effect on IgE and eosinophils by inhibiting B-cell proliferation and by downregulating eosinophil chemotaxis and Th2-associated chemokine activity [34].

IL-13 is involved in BP-associated itch through direct stimulation of peripheral nerve fibers and indirectly by recruiting IL-31-secreting eosinophils to the site of skin lesions. Indeed, dupilumab may improve pruritus by a direct effect on IL-4 and IL-13 and indirectly through the downregulation of IL-31 secretion by eosinophils [35].

The FDA approved dupilumab in 2017 for the treatment of moderate-to-severe atopic dermatitis, and it has been studied in relation to other type 2 inflammatory diseases, including BP [36].

Currently, there is an ongoing phase 2/3 randomized double-blind placebo-controlled trial for dupilumab administration in BP (NCT04206553) [37].

In 2018, Kaye et al. firstly described a case of recalcitrant BP successfully treated with dupilumab [38]. Afterwards, the use of dupilumab in BP treatment was reported in several case reports with encouraging results.

An interesting case report illustrates the management of a 70-year-old man with BP poorly responding to standard treatments including dapsone and corticosteroids. Omalizumab was associated to his conventional therapy (mycophenolate mofetil and high-potency corticosteroids), with the persistence of mild itch and transient skin lesions. Therefore, dupilumab was added to his treatment, obtaining clinical remission at the 7-month-follow up visit. Both mycophenolate mofetil and high-potency corticosteroids were stopped, and the patient remained in complete remission at the 3-month-follow up visit [39].

Seidman et al. described the case of a man affected by BP and type 2 diabetes mellitus. The patient was firstly treated with prednisone, but his pruritus and diabetes failed to be sufficiently controlled. He then started a trial with omalizumab as an alternative corticosteroid-sparing agent, but after 6 months of treatment, he experienced a disease flare. Therefore, the patient was started on dupilumab and exhibited great pruritus improvement and resolution of his BP lesions [40].

Abdat et al. reported a multicenter case series of 13 patients from five academic centers receiving dupilumab for refractory BP. The results showed that 92.3% (12 of 13 patients) presented clinical improvement: 53.8% (7 of 13) reached total disease clearance, defined as healing of all previously identified lesions with no further blister formation; 34% achieved satisfactory disease control in pruritus or bullae, defined as a patient desiring to stay on the medication. Only one patient did not respond.

Dupilumab was administered at the dosing regimen approved for atopic dermatitis: 600 mg SC initially, followed by 300 mg SC every other week. However, 42.9% of patients (3 of 7) needed a maintenance dose more frequently than every other week, suggesting that higher doses of dupilumab may be necessary for some patients.

Finally, 3 of 13 patients achieved disease clearance while receiving dupilumab in conjunction with a conventional therapy such as methotrexate, prednisone, and intravenous immunoglobulins [41].

Zhang et al. retrospectively reviewed 24 patients with moderate-to-severe BP: the first group consisted of 8 patients treated with dupilumab plus azathioprine and methylprednisolone, while the control group comprised 16 conventionally treated patients (methylprednisolone and azathioprine). Dupilumab in addition to methylprednisolone and azathioprine resulted in better control of disease progression, assessed as the mean time to stop new blister formation (8 days vs. 12 days). Moreover, the addition of dupilumab to conventional therapy allowed the acceleration of tapering of the glucocorticoid dose, evaluated as the mean time to reduce the systemic glucocorticoids to a conventional minimal dose of methylprednisolone at 0.08 mg/kg/day (121.5 vs. 148.5 days) [42].

All the results described above are summarized in Table 2.

### 3.3. Omalizumab

IgE and eosinophils are strongly involved in the pathogenesis of BP, as the level of anti-BP180 IgE NC16a domain correlates with disease activity, based on the BPDAI score, as reported in two case-control studies [43]. Patients with high levels of anti-BP180 IgE had twice the extension in body surface area (BSA) compared to patients with only anti-BP180 IgG. A positive correlation was also observed between BSA and total IgE, IgG anti-BP180, and IgE anti-BP180. A positive correlation was also observed between disease remission and the decrease in total IgE, specific IgE, and specific IgG [44,45].

However, in patients with complete remission, defined as the absence of manifestations of disease being observed for at least 4 months, 24% had IgE at normal limits, 9.3% of patients had IgG negativation, and 81% had specific IgE negativation. Both total IgE and specific IgE were found to be elevated in 79% of the patients investigated [46,47].

Other studies reported variable data on the value of specific IgE in BP, differing from 22% to 100% [48,49,50,51,52,53,54,55]. This variation could also be attributable to the lack of standardization in methods used for detection with different sensitivities. Although many studies have focused on the detection of IgE antibodies against the BP180 NC16a domain, there are also anti-BP180 IgE against other domains and anti-BP230 IgE, whose main target has not been precisely identified but seems to belong to the C-terminal domain [56]. Specific anti-BP230 IgE was identified in 50% and 67% of cases, respectively, in two studies and in 100% of BP cases in another study [57,58,59]. The search for specific IgE antibodies seems unhelpful in diagnosis, as the addition of the search for anti-BP180 IgE to the search for anti-BP180 IgG increases sensitivity by only 2.2%.

Anti-BP180 IgE was also detected via direct immunofluorescence (DIF) in 41% of patients with bullous pemphigoid, and in 5% of cases the diagnosis was made just by the detection of IgE with DIF, with negativity for IgG and C3 [60,61].

Eosinophilia is variably found in 40–80% of patients with BP, with a mean value around 50%, as reported in a recent, large case-control study [62,63]. Elevation of serum levels of eosinophil cationic protein (ECP), major basic protein (MBP) and eosinophil-derived neurotoxin (EDN) has also been demonstrated. The histological presence in skin lesions and blister fluid of eosinophils, ECP, and EDN detected by ELISA but not MBP is responsible for the damage to the dermo-epidermal junction [64,65]. The presence of eosinophils in the areas affected by the inflammatory lesions of pemphigoid is favored by chemokines such as eotaxin, with higher-than-normal concentrations in both serum and blister fluids of IL-5 [34]. Metalloproteases such as MMP-9 (matrix metallopeptidase 9), eosinophilic granules, and EETs (eosinophil extracellular traps) are involved in DEJ (dermo-epidermal junction) damage [66,67,68]. Eosinophils would appear to be responsible for the damage due to the binding of anti-BP180 IgE to the FCeRI receptors expressed by mast cells and basophils but also by eosinophils themselves, with subsequent degranulation [69].

The study of eosinophils and IgE in the pathogenesis of BP resistant to canonical therapies represented the rationale for employment of omalizumab in the treatment of non-responsive BP.

Omalizumab (OMZ) is a monoclonal antibody that recognizes the Cε3 domain of IgE with high affinity, blocking the latter′s interaction with the specific FcεRI receptor, reducing not only serum IgE levels but also the level of eosinophilia [70]. OMZ also appears to result in a reduction of FcεRI receptors both at the level of lesions and at the level of circulating basophils [71,72]. It is currently approved for use in chronic idiopathic urticaria and asthma [73,74].

Numerous case reports and case series have been published on the use of OMZ in patients with BP. In most cases, total IgE and eosinophils were also measured. They were frequently found to be elevated at the start of therapy, with a rapid reduction in eosinophils after therapy and a slower, sometimes undetectable decrease in IgE, probably due to the immunocomplexes between the drug and IgE that have a higher clearance. The change in the titers of anti-BP180 IgG and anti-BP230 IgG antibodies following OMZ therapy is also variable, with values decreasing in some cases, albeit slowly, and values remaining unchanged in other cases [75,76,77,78].

In one of the largest case series, which involved 11 patients, no correlation between response to OMZ and eosinophil and IgE levels in the blood emerged [79]. Discordant data exist concerning the response to OMZ and the presence or absence of specific antibodies such as IgE anti-BP180 and/or IgE anti-BP230. Two studies seem to demonstrate a clinical response even in the absence of specific IgE, while one case series showed that the two patients who failed to respond to OMZ were those with no detectable serum levels IgE, although three other patients without specific IgE showed a response to the treatment [80,81,82].

Specific IgG antibody levels tend to remain stable or drop very slowly over the course of months [83].

In the various reported cases, therapy was administered differently, both in terms of frequency of administration and dosage. Some studies were based on the asthma dosing nomogram, while others used the posology of chronic urticaria, leading to dosages ranging from 375 mg E2W to 300 mg E8W. In some case reports the dosing interval was varied from the initial scheme, adjusting the dose interval (lengthening or shortening) according to disease course [84,85].

The time of disease recurrence after OMZ cessation is variable, with some studies reporting a recurrence within a few weeks, and others reporting a complete remission of the disease even after several months [28,86].

However, most patients were already on other immunosuppressive therapies, including corticosteroids, and the time of disease recurrence may also depend on whether the other therapies were completely or partially discontinued, as well as on the duration of OMZ treatment before its discontinuation. In one case report, after failure with rituximab and IVIg, a complete remission was achieved with four doses of OMZ E3W, maintained over 5 months of FU with 5 mg/day of prednisone 85. The case series of Alexandre et al. reported a very low relapse rate, with a complete response maintained in 1/3 of the patients with minimal therapy after OMZ discontinuation, probably due to tapering of the OMZ occurring only after the very slow discontinuation of corticosteroids, while seven other patients maintained remission by remaining on OMZ [87].

OMZ allowed for the tapering of corticosteroids in all responding or partially responding patients, reducing the long-term complications related to corticosteroid therapy. No side effects were observed in most studies.

In a few reported cases with side effects, a clear cause–effect relationship between side effects and therapy was not established easily, especially because most of the patients were elderly and frail, and, in some cases, were coming from long treatments with immunosuppressive drugs without clinical benefit. In one non-responding patient, post-infusion flares occurred. One death was due to flu followed by bacterial superinfection. [88] Two patients died due to myocardial infarctions with no causality from OMZ, as the two patients were at high cardiovascular risk. Three deaths of patients between 84 and 90 years of age with comorbidities were observed out of 13 cases: two during treatment with OMZ, for ab ingestis pneumonia and COVID-19 pneumonia, respectively, and another from renal failure 3 months after discontinuation of OMZ. Another case series reports high levels of liver enzymes and thrombocytopenia. Two patients developed a mild flu-like illness after injection, which was easily managed with paracetamol and disappeared from the sixth injection onwards [50,78,87].

Mangin et al. reported the occurrence of acquired hemophilia A in two patients after OMZ administration. In one case the hemorrhagic blister resolved spontaneously within few days, allowing OMZ administration at a dosage of 600 mg to be continued. In the other case, the onset of acquired hemophilia A led to the discontinuation of OMZ and treatment with mycophenolate mofetil, even though the causal relationship is questionable.

Appreciable improvements have been reported in some cases of BP with mucosal (especially laryngeal) involvement that were refractory to other therapies, with longer disease-free times [89].

Benefit was also observed in a 5-month-old patient with eosinophilia and elevated total IgE whose illness was unresponsive to other therapies, without adverse events and with disease control in 25 days [81].

Sinha et al. reported that in patients already on prednisolone and azathioprine therapy with poor disease control, a single injection of OMZ at a dose of 450 mg resulted in disease remission while maintaining the other two drugs as therapy [90].

In conclusion, although the role of eosinophils and IgE in the pathogenesis of BP is not fully elucidated, scientific evidence seems to support the possible systematic use of OMZ as a therapy in refractory cases. There is currently an ongoing open-label, single-group trial testing the efficacy of rituximab combined with the administration of 300 mg omalizumab every 2 weeks (NCT04128176) [82].

All the results described above are summarized in Table 3.

### 3.4. Other Therapies

#### 3.4.1. Complement System Inhibitors

Complement activation is widely recognized as a necessary step in the pathogenesis of BP. Direct immunofluorescence (DIF) of perilesional skin shows linear deposits of C3, as well as IgG in most cases. Moreover, complement components and activation fragments including C1, C3, C3d, P, C5, and membrane attack complex (MAC) have been found at the basal membrane and blister fluid in BP [91,92].

C5a represents a strong chemokine for granulocytes and mediates the initiation of tissue inflammation in response to IgG autoantibody deposition. Leukotriene B4 (LTB4), stimulated by C5a, and its receptor BLT1 play a critical role in neutrophil recruitment at the dermal–epidermal junction. Elevated levels of C5a and LTB4 have also been detected in the blister fluid of patients with BP [93,94,95].

○Nomacopan (rVA576, a recombinant small protein, formerly known as coversin) is a complement inhibitor with activity against both C5 and LTB4. In a phase 2 nonrandomized, controlled trial (NCT04035733), seven of nine patients with mild-to-moderate new-onset or relapsing BP treated with nomacopan showed no treatment-related AEs (primary endpoint) and a significant decrease in disease activity and improvement of quality of life (secondary endpoint). A randomized, double blind, placebo-controlled clinical trial is expected to enroll 148 participants to evaluate nomacopan’s efficacy regarding the primary endpoint (NCT05061771) [96].○Avdoralimab, a specific anti-C5aR1 monoclonal antibody, has already shown a good safety profile in the treatment of solid tumors and rheumatoid arthritis. While C5aR1 mediates anti-BP180 IgG-induced pathogenicity, C5aR2 has a protective effect. Investigators hypothesize that avdoralimab might be a safe and effective treatment in BP patients: an open label, randomized, parallel-group phase 2 clinical trial (NCT04563923) is expected to enroll 40 patients to evaluate the efficacy of avdoralimab in addition to superpotent topical steroids. Complete clinical remission is defined as the primary endpoint [97,98].○Sutimlimab (BIVV009, formerly known as TNT009) is a humanized IgG4 monoclonal antibody that targets the C1s component of complement and thus inhibits leukocyte chemoattraction in BP. A phase 1 trial (NCT02502903) was conducted on 122 patients to study the safety, tolerability, and activity of sutimlimab in healthy patients and patients with complement-mediated disorders. Sutimlimab had a good safety profile and predictable and consistent pharmacokinetics and pharmacodynamics in healthy volunteers. In 10 patients with active or past BP, the classical complement pathway was blocked, and C3c deposition along the DEJ was partially or completely abrogated in 4 of 5 patients.

The FDA designated sutimlimab as an orphan drug for BP in 2017 to promote future clinical investigations [99,100].

#### 3.4.2. Eotaxin-1 (CCL-11)

Eotaxin-1 (CCL-11) is a crucial chemokine involved in the recruitment of Th2 effector and eosinophilic cells into BP lesions. It is expressed at high levels in blister fluid, in perilesional skin, and in serum, demonstrating a positive correlation with tissue eosinophilia [63].

Bertilimumab is a fully human monoclonal antibody that targets eotaxin-1, impairing eosinophil migration to the skin. It has been evaluated in a completed phase 2 clinical trial on nine patients with moderate-to-extensive BP (NCT02226146), demonstrating an 81% reduction in disease severity. The drug was well tolerated, and no significant adverse events were reported [101].

#### 3.4.3. Dimethyl Fumarate

Dimethyl fumarate (DMF), a derivative of the Krebs cycle intermediate fumarate, is an oral immunomodulator drug approved for the treatment of multiple sclerosis and moderate-to-severe plaque psoriasis. DMF acts by downregulating aerobic glycolysis in activated myeloid and lymphoid cells, by inhibiting the infiltration of neutrophils and monocytes into the skin, and by activating a nuclear factor (NRF2) involved in protecting cells from oxidative damage. DMF has recently been shown to be effective in treating the BP-like acquired epidermolysis bullosa (EBA) in a murine model [102].

In the literature is reported the case of a 69-year-old woman affected by multiple sclerosis for 35 years and with a diagnosis of bullous pemphigoid who was treated successfully with DMF, which led to a complete remission of BP lesions after 1 year of therapy [103].

#### 3.4.4. IL-5 Inhibitors

IL-5 is a chemoattractant, Th2-induced cytokine that promotes the recruitment and activation of eosinophils. In BP skin lesions, it is detectable, and its level correlates positively with disease activity [104].

Mepolizumab is a humanized IgG1 kappa monoclonal antibody that binds to IL-5 and prevents its interaction with the eosinophil surface receptor. This monoclonal drug is used for the treatment of asthma. In a randomized, placebo-controlled phase 2 study (NCT01705795), 30 patients were enrolled for a short, 12-week period and treated with mepolizumab at a dosage of 750 mg four times for four months. The primary endpoint was the cumulative rate of relapse-free patients after initiation of therapy. Although mepolizumab significantly reduced peripheral blood eosinophil counts, skin eosinophil infiltration was not significantly affected. Thus, mepolizumab did not meet the primary endpoint for BP and was not studied further [105,106].

Benralizumab is another IL-5 inhibitor used in asthma with therapeutic prospects in BP. This humanized IgG1 k monoclonal antibody binds to the IL-5R α-subunit on eosinophils and basophils, blocking their differentiation and maturation. No BP-related studies have been completed, but multinational, randomized, double-blind, placebo-controlled phase 3 clinical trials (NCT04612790) on the topic are currently underway; 120 patients have been enrolled, and complete remission at 36 weeks is the primary endpoint [107].

#### 3.4.5. IL-17 and IL-23 Inhibitors

Evidence suggests a significant role of IL-17 and Il-23 in the inflammatory pathogenesis of BP. IL-17 is elevated in the serum, blister fluid, and perilesional skin of BP patients. All isoforms of IL-17 are more present in blister fluid than in serum. IL-17 receptors have a discordant role, as the IL-17RA and IL-17RC receptors are associated with increased disease activity, while IL-17RB has a protective role in BP.

In addition, IL-17 promotes the secretion of MMP-9 and neutrophil elastase, both of which are involved in the blister-formation process. IL-23 is a cytokine that promotes the expression of IL-17 and the direct secretion of MMP-9. Based on this evidence, targeting the IL-23/IL-17 axis seems to be a good therapeutic option in BP. Regarding IL-17A inhibition in BP, a phase 2 clinical trial (NCT03099538) evaluated the activity of ixekizumab; four patients were enrolled, and cessation of blister formation was the primary endpoint. Subcutaneous ixekizumab at a dosage of 160 mg was administered at week 0, with 80 mg administered at weeks 2, 4, 6, 8, 10, and 12. This study was interrupted for lack of benefit [108].

Ustekinumab, a p40 subunit inhibitor of IL-12/23, seems to have a therapeutic role in BP. A single case report describes the successful treatment of recalcitrant BP with ustekinumab [109]. However, paradoxical development of BP has been reported during treatment of psoriasis with ustekinumab. A phase 2 open-label, single-arm study (NCT04117932) is currently underway, and patients can still be recruited; 18 patients have been enrolled. The aim of the study is to evaluate efficacy of ustekinumab in association with topical corticosteroids in BP patients during a period of 8 weeks. Complete remission from disease is considered to be the primary endpoint [110,111,112].

Specific IL-23 inhibitors (risankizumab, guselkumab, tildrakizumab, and mirankizumab) could have a hypothetical benefit in refractory BP. Regarding tildrakizumab, an open-label, single-arm phase 1 study (NCT04465292) is underway; 16 patients were enrolled, and recruitment was closed. Change in disease severity was defined as the primary endpoint [113].

#### 3.4.6. Inflammasome Inhibition

The NLRP3 inflammasome is a multimeric protein complex that controls caspase-1 and promotes the release of proinflammatory cytokines [114].

Recently, Fang et al. demonstrated increased expression of NRPL3 inflammasome components in peripheral blood mononuclear cells of BP patients. In this study, activation of the inflammasome by proinflammatory cytokines (IL-1ß, IL-17, and IL-23) was observed [115,116].

Although NLRP3 seems to participate in inflammatory pathogenesis, pharmacological modulation could be interesting for the treatment of BP [10].

#### 3.4.7. AC-203

AC-203 is an inflammasome and IL-1β modulator. It is used in topical ointment formulation. In vitro in keratinocyte lines treated with anti-BP180 IgG, the concomitant application of AC-203 was demonstrated to reduce the secretion of IL-6 and IL-8 [117]. A randomized, open-label phase 2 clinical trial comparing the effects of AC-203 ointment with clobetasol 0.05% ointment was recently completed (NCT03286582). Ten patients were enrolled, and the primary outcome was the incidence of adverse events during the treatment period. No results from this study are currently available [118].

#### 3.4.8. Ligelizumab

Ligelizumab (QGE031) is a humanized, high-affinity, second-generation anti-IgE monoclonal antibody. Compared with omalizumab, ligelizumab demonstrated greater inhibition of IgE binding to FcεRI, basophil activation, and IgE production by B-cells, as well as lower inhibition of IgE binding to CD23 [119,120].

Recently, Gasser et al. showed that the two anti-IgE antibodies exhibit different inhibition profiles and also show different ability in inhibiting IgE interactions with FcεRI and CD23 [121].

A randomized, placebo-controlled phase 2 clinical trial evaluating QGE031 in patients with BP was conducted (NCT01688882). After enrollment of 20 patients and administration of QGE031 at a dosage of 240 mg Q2W s.c., the predefined efficacy criteria were not met.

For some authors, the failure of this study may be due to the choice of primary endpoint, the selection of patients that did not consider the level of IgE, and the drug dose used [122].

All the results described above are summarized in Table 4.

## 4. Discussion 

Bullous pemphigoid is nowadays a therapeutic challenge, as the only validated therapies consist of steroids and steroid-sparing immunosuppressants, whose rate of efficacy is counterbalanced by their low safety profile over long-term use [123,124].

The latest discoveries on the pathogenesis of BP, however, have given an impetus for further research aiming to identify new target treatments for refractory cases, with the hope of guaranteeing long-term-effective and safe treatments for patients. As described earlier, the choice of possible targets ranges from CD20+ lymphocytes with rituximab to the Th2 axis using dupilumab and omalizumab or the IL-17/IL-23 axis to the inhibition of certain molecules of the complement system or the inflammasome. 

Rituximab has already entered into the therapeutic armamentarium of dermatologists, but only for treatment of pemphigus vulgaris, although preliminary evidence has demonstrated high rates of remission, steroid-sparing activity, and an acceptable safety profile in patients with severe BP or disease refractory to conventional therapies.

Several data in the literature highlight the use of dupilumab as an implemented therapy, owing to its high safety profile. Its use has been shown to be promising for the treatment of fragile patients in whom attention to maintaining immunological surveillance is mandatory (e.g., neoplastic, TBC-infected patients), also due to the scarce drug interaction with other biologics (e.g., nivolumab). Dupilumab has also been demonstrated to be effective in controlling symptoms, completely or greatly reversing pruritus in most treated cases. It represents one of the most promising treatments to come, and a phase 2/3 randomized, double-blind, placebo-controlled trial of dupilumab administration in BP is currently ongoing.

Omalizumab is increasingly recognized as a potential alternative agent in the treatment of BP, and the evidence to support this practice is increasing. 

The positive treatment responses reported in the literature reinforce the potential value of wider, prospective studies to confirm its efficacy and safety for the treatment of BP.

Although additional studies involving more patients will be necessary to evaluate the benefit/risk profile of complement system inhibitors, available data suggest that selective targeting of the classical complement pathway offers a promising approach to BP therapy.

The role of eotaxin in recruiting activated cells at inflammatory sites during BP supports its role as therapeutic target, and results from a pivotal phase 2 clinical trial on nine patients with moderate-to-severe BP treated with bertilimumab (fully human monoclonal antibody that targets eotaxin-1) make research on this molecular target a promising challenge. Eosinophil infiltration and eosinophilic spongiosis are prominent features in BP, and several data from the literature support the crucial pathogenic role of eosinophils in BP. IL-5, eotaxin, and eosinophil colony-stimulating factor are over-expressed in blister fluid; eosinophils releasing metalloprotease-9 and eosinophil degranulation proteins are found at the dermo-epidermal junction in lesional skin of BP patients; IL-5-activated eosinophils move to the dermo-epidermal junction in the presence of BP serum; and eosinophils have been demonstrated to be crucial in anti-BP180 IgE-mediated skin blistering [64,65,125]. Moreover, it is likely that eosinophils contribute to the pathogenesis of BP in several other not-yet-explored ways, owing to the current limits in the understanding of the biology of eosinophils. For all these reasons, experimental treatments aiming to target “eosinophilic pressure” in BP are looked upon with hope, such as bertilimumab, benralizumab, and mepolizumab, for which phase 2–3 trials are ongoing.

Several pieces of evidence suggest a significant role of the IL-17/IL-23 axis in BP pathogenesis: increased numbers of IL-17A+ CD4+ lymphocytes have been identified in peripheral blood of BP patients, and genes codifying the IL-17/23 axis have been shown to be upregulated in patients with BP [126,127].

The inhibition of dermal–epidermal separation in cryosections of human skin incubated with anti-BP180 IgG and subsequently with anti-IL-17A IgG-treated leukocytes has been demonstrated; a close correlation between serum IL-17A and IL23 levels and disease activity in a mouse model of BP has been reported; IL17A-deficient mice have been demonstrated to be protected against autoantibody-induced BP, and pharmacological inhibition of lL-17A seems to reduce the induction of BP in mice. 

Giving a role to the IL-17/IL-23 axis in the pathogenesis of PB, great expectation is placed on the ongoing studies on inhibitors of the IL-17/IL-23 axis in BP, owing also to the great confidence of use derived from their frequent use in psoriasis.

The putative interaction between IgE and eosinophils is a primary focus in current studies aimed at understanding the key components of BP, and the interest in therapeutics targeting IgE is increasing.

The results reported in this review have several limitations. Owing to the review’s retrospective nature, control groups for each reported treatment are lacking, failing to allow a direct comparative assessment of the effects of the reported treatments on patient outcomes; the sample size for each treatment is small; all results have been reported as single-academic-institution experiences; laboratory tests such as DIF, IFF, ELISA, IgE, and eosinophil levels were not systematically available.

Although research in the field of molecular target therapies for BP is currently in ferment, it should be noted that all of these treatments are off-label, and their use should be carefully considered by dermatologists and therefore selectively restricted to refractory or complicated cases for which long-term treatment with corticosteroid or immunosuppressive drug therapies is inappropriate. 

## Figures and Tables

**Figure 1 biomedicines-10-02844-f001:**
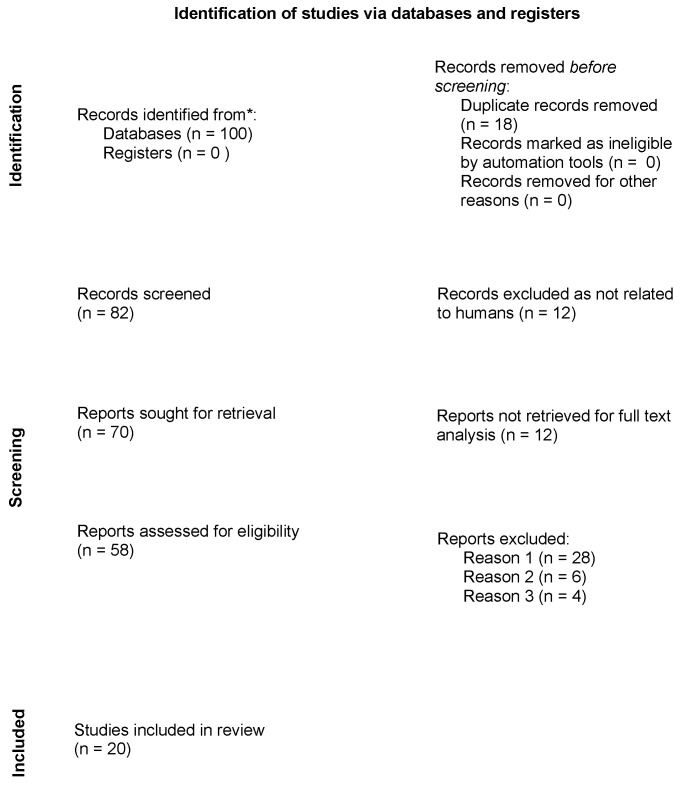
Preferred Reporting items for Systematic Reviews and Meta-Analysis (PRISMA) on novel therapeutic approach to bullous pemphigoid.

**Table 1 biomedicines-10-02844-t001:** Studies summarizing rituximab use in BP.

Authors	Drug	No. of Patients	Dose	Efficacy	Safety	Associated Therapy
Schmidt et al., (2015) [20]	Rituximab	13	500 mg weekly for 4 weeks	>90%	infections	PDSN
Polansky et al., (2019) [21]	Rituximab	20	1 g repeated in 2 weeks or 375 mg/m^2^ weekly for 4 weeks	75%	infections	PDSN, MFM, AZA, MTX
Tovanabutra et al., (2019) [22]	Rituximab	38	1 g repeated in 2 weeks or 375 mg/m^2^ weekly for 4 weeks	76%	-	PDN
Lamberts et al., (2018) [23]	Rituximab	28	500 or 1000 mg on days 1 and 15	67,9	-	-
Ahmed et al., (2015) [26]	Rituximab	12	4 weekly infusions of 375 mg/m^2^	100%	-	IVIg
Kremer et al., (2018) [28]	Rituximab	62	initial dose of 375 mg/m^2^ every 1–4 weeks to 500 mg weekly for 2 weeks	85%	infections, anemia, neutropenia, syndrome of inappropriate antidiuretic hormone secretion (SIADH), drug fever, acute pruritus, peripheral arterial occlusive disease and tachycardia.	-

PDSN = prednisolone; MFM = mycophenolate mofetil; MTX = methotrexate; PDN = prednisone; IVIg = intravenous immunoglobulin; AZA = azathioprine.

**Table 2 biomedicines-10-02844-t002:** Studies summarizing dupilumab use in BP.

Author	Drug	No. of Patients	Dose	Primary Endpoint	Safety	Associated Therapies	Phase of Study
NCT04206553 [37]	Dupilumab	98 (estimated)	Loading dose administered SC, followed by once every 2 weeks (Q2W)	Proportion of patients achieving sustained remission	-	-	Phase 2/3
Kaye A. et al., (2018) [38]	Dupilumab	1	600 mg week 0, 300 mg every other week	Complete remission	NA	No	Case report
Seyed Jafari S. et al., (2020) [39]	Dupilumab	1	600 mg week 0, 300 mg every other week	Complete remission	NA	Omalizumab, MFM, TCS	Case report
Seidman S. et al., (2019) [40]	Dupilumab	1	600 mg week 0, 300 mg every other week	Improved pruritus, complete remission	NA	PDN, MFM, DXC, nicotinamide, TCS	Case report
Abdat R. et al., (2020) [41]	Dupilumab	13	600 mg week 0, 300 mg every other week; 600 mg week 0, 300 mg weekly. 600 mg week 0, 300 mg every 12 days;	7/13 Disease clearance 5/13 Satisfactory disease control 1/13 No response	NA	MTX, PDN, IVIg	Case series
Zhang Y et al., (2021) [42]	Dupilumab	8	600 mg week 0, 300 mg every other week	Complete remission (62.5%)	NA	AZA, MPDSN	Comparative Study

MPDSN = metilprednisolone; MFM = mycophenolate mofetil; MTX = methotrexate; PDN = prednisone; IVIg = intravenous immunoglobulin; AZA = azathioprine; DXC = doxycycline; TCS = topical corticosteroids.

**Table 3 biomedicines-10-02844-t003:** Studies summarizing omalizumab use in BP.

Author	Drug	No. of Patients	Dose	Efficacy	Adverse Effects	Associated Therapies	Failed Treatments
Menzinger S. et al., (2018) [49]	OMZ	1	300 mg monthly	Disease was completely controlled (complete remission, no blister and pruritus)	None	CLB tapered and stopped	
Dufour C. et al., (2012) [81]	OMZ	1	100 mg E2W and E4W	Disease control	Not reported	Not clear	PDSN, MPDSN, topical betamethasone, DPN, AZA
London V.A. et al., (2012) [76]	OMZ	1	300 mg E4-8W	Complete remission (no blister and no pruritus)	None	AZA PDN	PDN, AZA, MFM, MPDSN, CFM
Pinar I.U. et al., (2017) [87]	OMZ	11	300 mg E2W-E4W to E8W.	6 complete clinical responses, 1 partial response, 4 N/A (not applicable).	Elevated liver enzymes, trombocytopenia, 2 myocardial infarctions not directly related to OMZ therapy	MPDSN, AZA, CLB, PDSN.	
Balakirski G. et al., (2016) [83]	OMZ	2	300 mg E4W to 300 mg E3W; 300 mg E3W	Free of pruritus and few isolated blisters; almost free of symptoms	None	PDSN; PDSN	PDSN + AZA; PDSN
Fairley J.A. et al., (2009) [48]	OMZ	1	300 mg E2W	Small amount of residual disease	N/A	N/A	PDN, AZA, minocycline.
Yu K.K. et al., (2014) [79]	OMZ	6	375 mg E2W; 300 mg E8W	2 disease-free, 3 symptom-free, 1 N/A caused by exacerbated COPD	COPD exacerbation related to termination of PDN; epigastric pain and mild elevation of liver enzymes (that responded to decrease of AZA)	PDN, AZA; PDN; PDN, AZA	PDN, AZA, minocycline; PDN niacinammide DXC; PDN; PDN, plasmapheresis, CFM, AZA; PDN; PDSN, AZA, plasmapheresis
Gönül M. et al., (2016) [77]	OMZ	1	300 mg E4W	Complete remission	Thrombocytopenia	PDSN	CLB, tetracycline, PDSN, DPN

MPDSN = metilprednisolone; PDSN = prednisolone; DPN = dapsone; CFM = cyclophosphamide; MFM = mycophenolate mofetil; MTX = methotrexate; PDN = prednisone; IVIg = intravenous immunoglobulin; AZA = azathioprine; DXC = doxycycline; TCS = topical corticosteroids; CLB = clobetasol propionate.

**Table 4 biomedicines-10-02844-t004:** Studies summarizing miscellaneous treatments for BP.

Author	DRUGS	N° Patients	dose	Primary Endpoint	Safety	Associated Therapies	Phase of Study
Pavord ID et al., (2012) [105] Simon D et al. (2020) [106]	Mepolizumab	30	750 mg four times for four months	Cumulative rate of relapse-free patients after initiation of therapy	No mepolizumab-related adverse events	No	Phase 3
FitzGerald JM et al., (2020) [107]	Benralizumab	120	Subcutaneously (SC) loading dose followed by repeat dosing	Complete remission at 36 weeks	NA	OCS	Phase 3
Arm JP et al., (2014) [119]	Ligelizumab	20	240 mg Q2W s.c.	Number of Patients That Had a Clinical Global Assessment of Change (CGA-C) Responder Rate by Week 12			Phase 2
NCT03099538 (2017) [108]	Ixekizumab	4	SC Ixekizumab 160 at week 0, 80 mg at weeks 2, 4, 6, 8, 10, 12 weeks	Cessation of blister formation	NA	No	Phase 2
NCT04117932 (2019) [112]	Ustekinumab	18	SC Ustekinumab 90 mg at weeks 0, 4, 16	Complete remission	NA	Superpotent TCS	Phase 2
NCT04465292 (2020) [113]	Tildrakizumab	16	SC Tildrakizumab 100 mg at weeks 0, 4 e 16	Change in disease severity	NA	No	Phase 1
Sadik CD et al., (2020) [96]	Nomacopan	9	SC Nomacopan 90 mg at day 1 30 mg daily until day 42	Incidence of grade 3, 4 and 5 adverse events	NA	No	Phase 2
Karsten CM et al., (2018) [98]	Avdoralimab	40	3 SC injections of avdoralimab every week for 12 weeks	Complete clinical remission at 3 months		CLB	Phase 2 underway
Bartko, Johann et al., (2022) [99]	Sutimlimab	10	Test dose of 10 mg/kg, followed by 4 weekly doses of 60 mg/kg EV	Drug-related Adverse Event for 6 weeks	NA	No	Phase 1
Fiorino, A.S et al., (2019) [101]	Bertilimumab	11	Intravenous 10 mg/kg, 3 doses biweekly	Safety endpoints		PDN 30 mg	Phase 2
Bilgic-Temel A et al., (2019) [103]	Dimethyl fumarate	1	120 mg/BD for 7 days and then increased to 240 mg/BD		NA	PDSN 25mg/day DXC 100 mg twice per day (BD), and nicotinamide 500 mg/BD	Case report
Topical therapies							
NCT03286582 (2017) [118]	AC-203	10	ointment applied twice a day	Incidence of adverse events during the treatment period	NA	No	Phase 2

PDSN = prednisolone; MFM = mycophenolate mofetil; PDN = prednisone; DXC = doxycycline; OCS = oral corticosteroids; TCS = topical corticosteroids CLB = clobetasol propionate.

## Data Availability

The data that support the findings of this study are available from the corresponding author upon reasonable request.

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
