# Peer review of "Bullous Pemphygoid and Novel Therapeutic Approaches"

_biomedicines, 2022, doi:10.3390/biomedicines10112844_

Round 1

Reviewer 1 Report

Dear Authors, the manuscript, although well structured, has some errors. Furthermore, some periods are completely transcribed with some changes from other works cited in bibliography. Below I point out some of the problems

1)In the rituximab paragraph there is an important error: from the line 217 to line 222
the phrases are referred to pemphigus not to bullous pemphigoid!! Therefore the whole
period must be deleted as well as the relative bibliography.
2)In the same paragraph the phrases between the 207 and 216 lines are completely copied with
some changes from the abstract of bibliographic reference number 28:these changes are inappropriate
and alter what is expressed in the abstract!
3)Use always BP not Bp

Author Response

Dear reviewer,
I have updated the text according to the indications given:

  1. we deleted the period and the relative bibliography
  2. we have changed the content of the paragraph
  3. all Bp's have been changed to BP

thanks for the suggestions

Reviewer 2 Report

The authors review bullous pemphigoid with a focus on novel therapeutics in recent 10 years and potential future therapies. The review is comprehensive and well-organized.

I have some comments.
1. For Figure 1, arrows are missing in the flow diagram of PRISMA. Please also explain, in step 2(screening), why 12 reports were not retrieved.
2. Line 65, the authors stated that this scoping review was based on the approach developed by Arksey and O'Malley. The citing reference 10 is not related to the approach.  The correct reference seems to be International Journal of Social Research Methodology 2005;8(1):19-32. Also, the approah developed by Arksey and O'Malley is different from PRISMA. Please clarify if the approach is consistent with PRISMA.
3. Line 606-617, the correct spelling is IL-17/IL-23 axis but not IL17-23 or Il17/23. 

Author Response

  1. For Figure 1, arrows are missing in the flow diagram of PRISMA (arrows have been reported now). Please also explain, in step 2(screening), why 12 reports were not retrieved (were not retrieved for full text analysis, this has been reported in revised version)
    2. Line 65, the authors stated that this scoping review was based on the approach developed by Arksey and O'Malley. The citing reference 10 is not related to the approach.  The correct reference seems to be International Journal of Social Research Methodology 2005;8(1):19-32. Also, the approah developed by Arksey and O'Malley is different from PRISMA. Please clarify if the approach is consistent with PRISMA. (we agree, and the reference has been changed accordingly)
    3. Line 606-617, the correct spelling is IL-17/IL-23 axis but not IL17-23 or Il17/23. (IL17/Il23 has been reported everywhere) 

Round 2

Reviewer 1 Report

At line 238 IL-13 must be changed with IL-31